# Disentangled Concept-Residual Models: Bridging the Interpretability–Performance Gap for Incomplete Concept Sets

**Renos Zabounidis**                                      *renosz@cs.cmu.edu*
*School of Computer Science, Carnegie Mellon Universtiy*

**Ini Oguntola**                                          *ioguntol@andrew.cmu.edu*
*School of Computer Science, Carnegie Mellon University*

**Konghao Zhao**                                          *konghaoz@usc.edu*
*Department of Computer Science, University of Southern California*

**Joseph Campbell**                                       *joecamp@purdue.edu*
*Department of Computer Science, Purdue University*

**Woojun Kim**                                            *woojunk@andrew.cmu.edu*
*School of Computer Science, Carnegie Mellon University*

**Simon Stepputtis**                                      *stepputtis@vt.edu*
*Department of Mechanical Engineering, Virginia Tech*

**Katia Sycara**                                          *sycara@andrew.cmu.edu*
*School of Computer Science, Carnegie Mellon University*

**Reviewed on OpenReview:** *https://openreview.net/forum?id=NKgNizwDa6*

## Abstract

Deploying AI in high-stakes settings requires models that are not only accurate but also interpretable and amenable to human oversight. Concept Bottleneck Models (CBMs) support these goals by structuring predictions around human-understandable concepts, enabling interpretability via steerability and post-hoc human oversight. However, CBMs rely on a 'complete' concept set, requiring practitioners to define and label enough concepts to match the predictive power of black-box models. To relax this requirement, prior work introduced residual connections that bypass the concept layer and recover information missing from an incomplete concept set. While effective in bridging the performance gap, these residuals can redundantly encode concept information, a phenomenon we term **concept-residual overlap**. In this work, we investigate the effects of concept-residual overlap and evaluate strategies to mitigate it. We (1) define metrics to quantify the extent of concept-residual overlap in CRMs; (2) introduce complementary metrics to evaluate how this overlap impacts interpretability by reducing steerability to concept interventions; and (3) present **Disentangled Concept-Residual Models (D-CRMs)**, a general class of CRMs designed to mitigate this issue. Within this class, we propose a novel disentanglement approach based on minimizing mutual information (MI). Using CelebA, CIFAR100, AA2, CUB, and OAI, we show that standard CRMs exhibit significant concept-residual overlap, and that reducing this overlap with MI-based D-CRMs restores the key properties of CBMs that make them interpretable, namely functional reliance on concepts and robust intervention behavior, without sacrificing predictive performance.

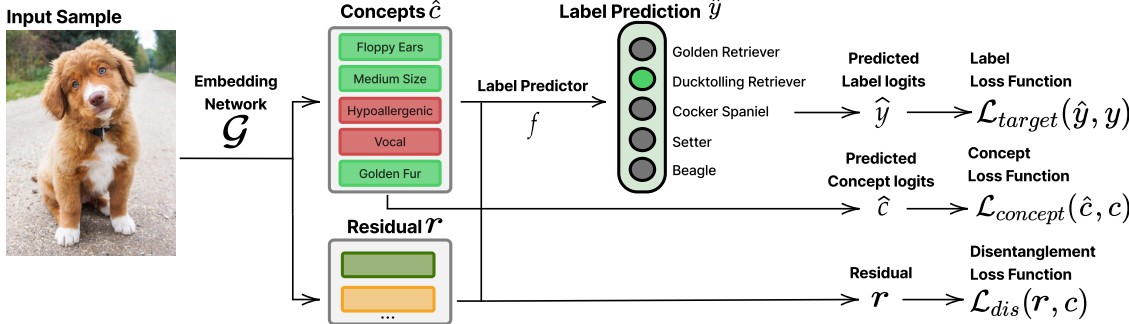

Figure 1: Overview of the Disentangled Concept–Residual Model (D-CRM). Given an input sample, the embedding network predicts concept logits and a residual representation. The predicted concepts are supervised using ground-truth labels and, together with the residual, are passed to the label predictor to produce class logits. A disentanglement objective encourages the residual to encode information complementary to the concepts, while a label loss ensures accurate classification.

## 1  Introduction

State-of-the-art deep learning models have achieved remarkable performance in complex tasks, surpassing humans in areas such as computer vision (Han et al., 2022), speech recognition (Radford et al., 2023), and strategic game playing (Silver et al., 2016; Brown & Sandholm, 2018). However, despite their high accuracy, the underlying decision-making processes of these models remain opaque to humans, leading to their characterization as *black boxes*. This opacity poses significant risks when deep learning is deployed in safety-critical applications, such as medical imaging (Salahuddin et al., 2022), where the issued diagnosis must be grounded in observed evidence.

Concept Bottleneck Models (CBMs) (Koh et al., 2020) enhance model interpretability by explicitly predicting intermediate, semantically meaningful concepts, which subsequently inform downstream predictions. These human-understandable concepts facilitate clear insights into model decisions, enabling users to trace predictions back to interpretable features. However, CBMs rely heavily on predefined comprehensive concept sets and sufficient labeled data, making them challenging to apply broadly in practice.

To address these limitations, Concept Residual Models (CRMs) (Yuksekgonul et al., 2023; Sawada & Nakamura, 2022; Zabounidis et al., 2023; Zhang et al., 2023; Espinosa Zarlenga et al., 2022) introduce an auxiliary residual layer that captures additional task-relevant information beyond the predefined concepts. While this improves downstream performance, it creates a challenge: residual layers can encode redundant concept information. This redundancy causes the downstream network to preferentially rely on the opaque residual representation rather than on the interpretable concept features. We term this phenomenon *concept-residual overlap*, and show it undermines intervention effectiveness and therefore interpretability originally sought by CBMs.

In this paper, we propose a comprehensive approach to tackle concept-residual overlap by enhancing disentanglement between concept and residual representations. We refer to this class of models as Disentangled Concept-Residual Models (D-CRMs). Central to our approach is a novel training objective based on minimizing an upper bound of mutual information (MI) between concept and residual representations, which addresses the root cause of reduced interpretability: models are no longer steerable via concepts when residuals redundantly encode concept information.

Our contributions can be summarized as follows:

• We propose mutual information as a measure for *concept-residual overlap* and analyze how redundant encoding of concept information across concept and residual pathways fundamentally impairs intervention effectiveness and therefore interpretability.

• We introduce and emphasize new model steerability metrics – random concept interventions and random residual interventions – that complement traditional performance evaluations, providing deeper insights into model steerability and residual reliance.

• We propose a novel mutual information minimization objective that reduces concept-residual overlap by 61-81% while maintaining performance within 3% of baseline CRMs, achieving 3-57× better concept importance preservation and intervention accuracy gains up to 16%.

• We empirically validate our approach across five diverse datasets, demonstrating that MI-based D-CRMs achieve near-complete overlap suppression on challenging datasets and maintain 98% counterfactual reasoning accuracy compared to 25% for standard CRMs. This significantly broadens the applicability and practical utility of interpretable concept-residual models.

## 2 Background

*Concept learning* approaches a prediction task from inputs $x \in X$ to outputs $y \in Y$ by introducing a low-dimensional intermediate representation $z \in Z$ that is in some way designed to correspond to semantically meaningful concepts (Lampert et al., 2009; Kumar et al., 2009; Kim et al., 2018; Koh et al., 2020; Chen et al., 2020). Typically these concepts are specified as a collection of ground truth values $c \in C$ representing human-interpretable properties of input (for example, the color of a bird's feathers), and are used during training to learn a suitable representation $z$.

For example, in the dog classification task shown in Figure 2, a model might be trained to first predict concepts like 'medium size' or 'brown fur' before predicting the final class, 'duck tolling retriever'. This structure is the foundation of Concept Bottleneck Models (CBMs).

### 2.1 Concept Bottleneck Models

CBMs aim to learn an intermediate concept representation $z = \hat{c}$ that directly predicts ground truth concept values $c$ (Koh et al., 2020). The model is decomposed into two components: a concept encoder $g : X \to C$ supervised on ground truth concept values, and a predictor $f : C \to Y$ supervised on the prediction task.

This approach has the advantage of being directly interpretable and straightforward to intervene over. For instance, in Figure 2, the model incorrectly predicts 'cocker spaniel' because it failed to recognize the 'medium size' concept. An expert can manually correct this single concept prediction, which should ideally correct the final prediction to 'duck tolling retriever'. This ability to intervene is a key advantage of CBMs.

Despite their benefits, vanilla CBMs suffer from two main limitations: concept leakage and the bottleneck assumption.

**Concept Leakage** Concept leakage occurs when extraneous information is encoded in the concept layer itself. This compromises interpretability and reduces intervention performance, as the downstream network relies on leaked features that interventions inadvertently remove. Mahinpei et al. were

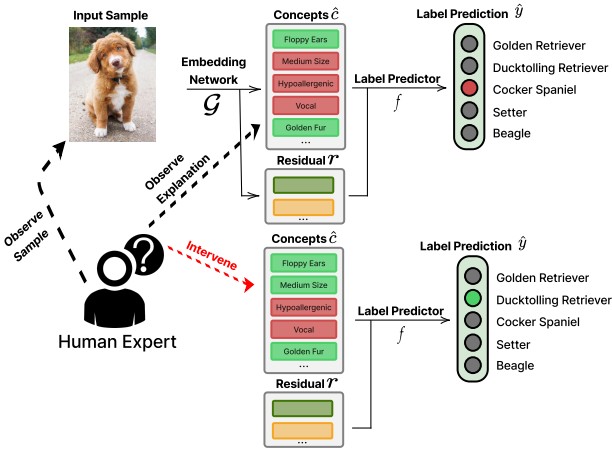

Figure 2: Demonstration of a test-time concept intervention on a mislabeled sample. The input image is a duck tolling retriever, but the model predicts cocker spaniel due to incorrectly predicting the "medium size" concept as false. Manually correcting the concept to true should ideally update the downstream prediction; however, if concept-residual overlap exists, the residual may still encode conflicting size information, preventing the prediction from updating. Our proposed D-CRMs mitigate this by enforcing disentanglement between concepts and residuals.

the first to name and empirically demonstrate this phenomenon in standard CBMs, showing how it degrades both interventionability and interpretability (Mahinpei et al., 2021; Margeloiu et al., 2021).

**Limited Capacity and the Bottleneck Assumption** CBMs rely on the bottleneck assumption, that the intermediate concept representation is sufficient to accurately predict the target label. This assumes that the provided set of ground truth concepts captures all information relevant to the downstream task. In our dog example, concepts like 'medium size' and 'brown fur' might not be enough to distinguish a 'duck tolling retriever' from a 'golden retriever'. The model might need information about 'fur texture' (e.g., wavy vs. straight), which may not be in the predefined concept set. When the concept set is incomplete, the model may struggle to make accurate predictions, which limits the practical utility of CBMs.

## 2.2   Concept-Residual Models

Concept-Residual Models (CRMs) are an extension of CBMs that mitigate the bottleneck assumption by additionally including a *residual* $r \in \mathbb{R}^l$ in the intermediate representation $z = (\hat{c}, r)$ to encode extra information relevant to the downstream prediction task (Sawada & Nakamura, 2022). The model is now decomposed into three components: a concept encoder $g : X \to C$, a residual encoder $h : X \to R$, and a predictor $f : C \times R \to Y$. In practice, many of the layers in $g$ and $h$ are shared. The residual can, in theory, learn the missing information about 'fur texture' to solve the classification task.

**Concept-Residual Overlap** By introducing a residual pathway, CRMs relax the bottleneck constraint. However, this flexibility introduces the risk of **concept-residual overlap**, where the same concept is redundantly encoded in both the concept layer $g(x)$ and the residual layer $h(x)$.

Returning to Figure 2, the residual, in addition to learning about 'fur texture', might also learn the 'medium size' concept. Now, when an expert intervenes and corrects the 'medium size' concept in the concept layer, the model might ignore this correction because the residual still contains the incorrect information about the dog's size. This creates ambiguity in the model's information flow and undermines the effectiveness of interventions. This ambiguity is the central problem we address in this paper. It undermines faithful attribution and intervention effectiveness, as modifying a concept may have little to no effect on the output if the residual retains the original, conflicting information.

## 3   Related Work

**CBM variants** Concept Bottleneck Models (CBMs), like those described by Koh et al. (2020), predict downstream tasks using an intermediary bottleneck layer trained with supervised losses. Bahadori & Heckerman (2020) extended CBMs to denoise the concepts. ProbCBM models uncertainty in concept prediction and provides explanations based on concept predictions and their corresponding uncertainty (Kim et al., 2023). Shin et al. (2023) develop various strategies for selecting which concepts maximally improve intervention effectiveness. Chauhan et al. (2023) similarly develop a method combining concept prediction uncertainty and influence of the concept on the final prediction to improve intervention effectiveness. CBMs offer interpretability advantages but are limited by the need for complete concept sets during training. The challenge lies in creating a fully representative set of concepts for a domain. When concepts are not expressive enough to capture all the needed information for the task, performance suffers.

**Label-Free CBMs** Recent advances in CBMs aim to reduce reliance on human-labeled concepts by automating concept discovery, reducing the assumption that concept labels exist for each dataset of interest. Label-Free CBM methods instead rely on foundation models to label concepts. Post-hoc CBMs (PCBMs) (Yuksekgonul et al., 2023) transform existing models into CBMs by leveraging concept activation vectors (CAVs) to discover and represent concepts. Label-Free CBMs (Oikarinen et al., 2023) take a different approach by learning a projection matrix that maximizes cosine similarity between the projection matrix output and CLIP scores, without requiring explicit concept labels. Both methods relax the requirement of labels, but still use a separate LLM to propose concept sets. Vision-Language-Guided CBMs (VLG-CBMs) (Srivastava et al., 2024) and Discover-then-Name CBMs (Rao et al., 2024) automate concept generation and naming using multimodal models and latent concept discovery, respectively.

Every label-free CBM method relies on foundation models to generate concept labels. As a result, these methods do not have access to ground-truth labels and therefore cannot evaluate or optimize for the ability to intervene on concept predictions. Since intervention is the primary mechanism through which concept leakage has been empirically demonstrated in CBMs, it remains unclear to what extent concept leakage compromises the interpretability of label-free approaches (Havasi et al., 2022). For example, Label-Free CBMs (Oikarinen et al., 2023) attempt to filter out uninterpretable concepts by requiring a minimum correlation threshold (40%) between the predicted concept activations and CLIP similarity scores for a corresponding concept prompt. However, this filtering does not eliminate the possibility of leakage: a concept that aligns with a CLIP embedding at 40% correlation may still represent a latent entanglement of multiple factors, or may not faithfully reflect the semantics of the intended concept at all. Since CLIP itself is not guaranteed to produce disentangled or interpretable concept representations, high correlation with its outputs may not be a sufficient condition for interpretability or faithful concept separation.

Our work addresses these limitations by integrating human-labeled concepts to ensure reliability, while utilizing additional residual capacity to capture information not represented in the labeled concepts. This hybrid approach combines the strengths of human oversight and automated discovery, enabling both interpretability and precision across a wide range of applications. By making stronger assumptions about label availability, we provide stronger guarantees regarding model transparency, disentanglement, and responsiveness to human intervention.

**Concept-Residual Approaches** Several works augment CBMs with residuals, with some recognizing that concept-residual overlap can degrade intervention effectiveness. Yüksekgönül et al. add a post-hoc residual predictor to CBMs, modeling $y = f(c(x)) + g(x)$, where $g(x)$ captures input-level information and can be removed for interpretability (Yuksekgonul et al., 2023). However, when $g(x)$ is active, it's unclear whether predictions still rely on concepts, and the additive form cannot capture nonlinear concept–residual interactions. Sawada et al. propose a similar residual augmentation but do not evaluate leakage (Sawada & Nakamura, 2022). Zabounidis et al. and Ismail et al. both demonstrate that orthogonality-based decorrelation improves intervention performance in CRMs and generative CBMs respectively, confirming that overlap undermines steerability (Zabounidis et al., 2023; Ismail et al., 2024). However, neither work quantifies overlap or systematically evaluates which measure of overlap best predicts intervention effectiveness. Zhang et al.'s DCBMs use mutual information to trace concept-label alignment but do not isolate residual contributions (Zhang et al., 2023). Oguntola et al. reduce intra-concept leakage via mutual information minimization in RL agents, improving rewards but not addressing interpretability under intervention (Oguntola et al., 2023).

Introducing Concept Embedding Models (CEMs), Espinosa Zarlenga et al. (2022; 2023) relax the strict bottleneck of CBMs by embedding each concept into a high-dimensional, continuous latent space. This improves representational flexibility and predictive performance, especially when the concept set is incomplete or noisy. However, the flexibility comes at the cost of interpretability: each concept embedding inherently mixes semantic content with residual information, making it difficult to reason about the causal role of individual concepts or to isolate their influence via counterfactual interventions. We provide further commentary in Appendix A.1.

Despite these efforts, no prior work systematically evaluates which measure of overlap best predicts intervention effectiveness. We address this gap by proposing MI as a principled measure, comparing it against decorrelation-based approaches, and demonstrating that MI-based disentanglement more effectively preserves steerability.

## 4 Method

We introduce **Disentangled Concept Residual Models (D-CRM)**, a unifying class of models designed to improve the interpretability and intervention fidelity of CRMs by minimizing concept-residual overlap. This framework encompasses a range of existing methods – such as *IterNorm* (Zabounidis et al., 2023) and *EYE* (Wang et al., 2022) – that aim to reduce redundancy between concept and residual representations. Within this general methodology, we propose two explicit techniques: (1) a *decorrelation loss* that penalizes linear dependencies between concept and residual representations, and (2) a *mutual information (MI) minimization objective* that enforces statistical independence. These techniques target different forms of

overlap, with MI minimization providing a more general approach by capturing both linear and nonlinear dependencies.

## 4.1 Mitigating CRM Overlap with Decorrelation

We introduce a targeted decorrelation loss to reduce overlap between concept and residual representations in CRMs. Unlike prior methods that enforce full orthogonality across concept and residual vectors, our approach selectively penalizes only the linear dependencies between the two, preserving any natural correlations within the concept space.

Formally, given a mini-batch of concept representations $C \in \mathbb{R}^{b \times k}$ and residual representations $R \in \mathbb{R}^{b \times d}$, we define the cross-covariance matrix as:

$$\mathbf{Cov}_{CR} = \frac{1}{b} C^\top R. \tag{1}$$

Our decorrelation loss penalizes the squared off-diagonal entries:

$$\mathcal{L}_{\mathrm{decorr}} = \sum_{i \neq j} (\mathbf{Cov}_{CR})_{ij}^2, \tag{2}$$

encouraging decorrelation between the concept and residual pathways while leaving intra-group correlations intact. This targeted approach contrasts with techniques such as ZCA whitening, used in prior work to promote concept interpretability or improve intervention performance (Chen et al., 2020; Zabounidis et al., 2023). Those methods decorrelate all dimensions-including between concepts-by applying a global transformation:

$$\mathbf{X}' = \mathbf{D}\Lambda^{-\frac{1}{2}}\mathbf{D}^T(\mathbf{X} - \mu_\mathbf{X}), \tag{3}$$

where $\mathbf{X}$ contains concatenated concept and residual vectors. While this enforces orthogonality, it also removes meaningful correlations among concept dimensions. By contrast, our loss focuses only on the interaction between concept and residual spaces. This makes it better suited for scenarios where concept relationships are semantically meaningful and should be preserved. Still, this approach only removes linear dependencies.

An alternative approach is *adversarial decorrelation*, which uses gradient reversal to prevent the residual from encoding concept information (Ganin & Lempitsky, 2015). A concept classifier $q_\phi : \mathcal{R} \to \mathcal{C}$ is trained to predict concepts from residuals, while the residual encoder $h$ is trained to maximize the classifier's loss via a gradient reversal layer. This adversarial game encourages the encoder to produce residuals that are uninformative about concepts. Unlike the decorrelation loss, adversarial decorrelation can capture nonlinear dependencies, though it requires careful tuning to balance the adversary and main objective.

## 4.2 Minimizing Mutual Information

To more generally quantify dependence between residual and concept representations, we use mutual information (MI) between residuals $r = h(x)$ and ground truth concepts $c$:

$$I(c; r) = H(c) - H(c \mid r) = D_{\mathrm{KL}}(p(c, r) \parallel p(c)\, p(r)), \tag{4}$$

where $H(\cdot)$ denotes entropy and $D_{\mathrm{KL}}$ is the Kullback-Leibler divergence. Mutual information measures how much knowing $r$ reduces uncertainty about $c$: higher values indicate stronger dependence, while $I(c; r) = 0$ corresponds to statistical independence, i.e., $p(c \mid r) = p(c)$, meaning the residual contains no information about the concepts. We estimate $I(c; r)$ using the Contrastive Log-ratio Upper Bound (CLUB) (Cheng et al., 2020), which learns a variational approximation $q_\theta(c \mid r)$ to the true conditional distribution. The estimator is trained by solving:

$$\min_\theta \ -\mathbb{E}_{p(c,r)} \left[ \log q_\theta(c \mid r) \right]. \tag{5}$$

Once trained, this estimator provides a differentiable approximation of mutual information. We then freeze $\theta$ and use the estimated $I(c; r)$ as a regularizer in the D-CRM training objective. Because $q_\theta$ is differentiable, gradients can propagate through it into the residual encoder $h$, enabling effective minimization of concept-residual overlap.

### 4.3  Semi-Independent Training Objective

D-CRMs are trained to balance accurate concept & target prediction with disentanglement between learned concepts and residuals. The model consists of: (1) a concept encoder $g : \mathcal{X} \rightarrow \hat{c}$, (2) a residual encoder $h : \mathcal{X} \rightarrow r$, and (3) a predictor $f : \mathcal{C} \times \mathcal{R} \rightarrow \hat{y}$, where $x \in \mathcal{X}$ is the input, $\hat{c} = g(x)$ are predicted concepts, $r = h(x)$ is the residual representation, and $\hat{y} = f(c, r)$ is the model's final prediction using ground truth concepts $c$ and learned residuals $r$.

**Gradient-blocked supervision.** Following Havasi et al. (2022), we supervise $g$ using ground truth concepts $c$, while preventing gradient flow from the target loss into $g$. This avoids concept leakage while maintaining alignment with $c$. The residual encoder $h$, however, remains fully trainable with respect to the target loss, enabling it to encode task-relevant information.

**Training losses.** The model is trained by minimizing a weighted combination of loss terms over the dataset distribution $\mathcal{D}$:

$$
\min_{g,h,f} \mathbb{E}_{(x,c,y)\sim\mathcal{D}} \left[ \alpha \underbrace{\mathcal{L}_{\text{concept}}(g(x), c)}_{\text{Concept supervision}} + \beta \underbrace{\mathcal{L}_{\text{target}}(f(c, h(x)), y)}_{\text{Prediction with fixed } c} + \gamma \underbrace{\mathcal{L}_{\text{disentangle}}(h(x), c)}_{\text{Disentanglement}} \right], \tag{6}
$$

where $\alpha, \beta, \gamma \in \mathbb{R}_{\geq 0}$ control the trade-off between concept fidelity, predictive performance, and representation disentanglement.

**Disentanglement via mutual information.** Because the CLUB estimate depends on a learned variational distribution $q_\theta(r \mid c)$, we must optimize both the model parameters and the estimator jointly. This results in a nested optimization problem: e. Specifically, we solve:

$$
\min_{g,h,f} \quad \alpha \, \mathcal{L}_{\text{concept}}(g(x), c) + \beta \, \mathcal{L}_{\text{target}}(f(c, h(x)), y) + \gamma \, I_{\text{CLUB}}(r, c; \theta^\star), \tag{7}
$$

where $\theta^\star$ is obtained by solving the estimator subproblem:

$$
\theta^\star = \arg\min_{\theta} \ -\mathbb{E}_{p(c,r)} \left[ \log q_\theta(c \mid r) \right]. \tag{8}
$$

In practice, we alternate updates by taking a single gradient step on $\theta$ to improve the mutual information estimate, and a single gradient step on the model parameters $(g, h, f)$ using $I_{\text{CLUB}}(r, c; \theta^\star)$ as a fixed, differentiable regularizer. This prevents the mutual information estimate from becoming stale as $r$ evolves, ensuring that gradients used to penalize overlap remain aligned with the current residual distribution.

## 5  Quantifying Concept-Residual Overlap and CRM Performance

While our mutual information formulation confirms the presence of concept-residual overlap, it does not indicate how this overlap affects the model's use of concept during target prediction. To address this, we introduce intervention-based metrics that measure the model's reliance on concepts versus residuals. These metrics help assess whether concepts meaningfully influence the model's output, or are ignored in favor of the residual as overlap increases.

### 5.1  Intervention-Based Metrics

Intervention-based metrics directly assess how changes to concept or residual inputs affect the predictions of the target model. We intervene on all concepts simultaneously, not individual concepts, which ensures that label-relevant concepts drive observable effects while irrelevant concepts contribute no signal.

**Positive Concept Interventions** We construct a modified input by setting all concepts to their ground-truth values: $c^{\text{intv}}(x) = c^{\text{true}}(x)$. Accuracy under positive interventions is defined as:

$$\text{ACC}_{\text{pos}} = \mathbb{E}_{(x,y) \sim \mathcal{D}} \left[ \mathbf{1} \left( f(c^{\text{intv}}(x), h(x)) = y \right) \right]. \tag{9}$$

High $\text{ACC}_{\text{pos}}$ indicates that concept updates are correctly propagated to model outputs. However, when predicted concepts are already correct, $\text{ACC}_{\text{pos}}$ provides limited information about concept reliance, motivating the need for random interventions.

**Random Concept Interventions** We randomly replace all concepts with values from other training examples:

$$\tilde{c}_i \sim \{c_i^{(j)} \mid j \in \mathcal{D}, \ c_i^{(j)} \neq c_i^{\text{true}}\}, \tag{10}$$

and measure:

$$\text{ACC}_{\text{neg}} = \mathbb{E}_{(x,y) \sim \mathcal{D}} \left[ \mathbf{1} \left( f(c^{\text{intv}}(x), h(x)) = y \right) \right]. \tag{11}$$

If the model truly relies on concepts, randomizing them should degrade performance even when the original predictions were correct. In CBMs, all predictive information flows through the concept layer, so randomizing concepts leads to random accuracy. For CRMs, where the target model has access to a residual, the degree of degradation reveals how much the CRM actually relies on concept representations for prediction. These two metrics are complementary: $\text{ACC}_{\text{pos}}$ measures sensitivity to correction when concepts are wrong, while $\text{ACC}_{\text{neg}}$ measures reliance when concepts are already correct.

**Residual Interventions** To test the model's reliance on the residual layer, we replace $h(x)$ with a residual from another sample:

$$\tilde{h}(x) \sim \{h(x^{(j)}) \mid j \in \mathcal{D}, \ j \neq i\}, \tag{12}$$

and define residual intervention accuracy:

$$\text{ACC}_{\text{res}} = \mathbb{E}_{(x,y) \sim \mathcal{D}} \left[ \mathbf{1} \left( f(c(x), \tilde{h}(x)) = y \right) \right]. \tag{13}$$

High $\text{ACC}_{\text{res}}$ indicates that concepts alone suffice for accurate target classification.

### 5.2 Feature Attribution with DeepLIFT

We assess the influence of each concept using DeepLIFT (Shrikumar et al., 2017), which assigns contribution scores $C_{\Delta c_i \Delta f_y}$ quantifying how the deviation of concept $c_i$ from a reference $c_i^{\text{ref}}$ contributes to the deviation of the model logit $f_y$ (for class $y$) from its reference output $f_y^{\text{ref}}$, such that:

$$\sum_{i=1}^{d_c} C_{\Delta c_i \Delta f_y} + \sum_{j=1}^{d_r} C_{\Delta h_j \Delta f_y} = \Delta f_y. \tag{14}$$

We average $C_{\Delta c_i \Delta f_y}$ over the validation set to estimate global concept importance. Baselines $c_i^{\text{ref}}$ are set to 0.5 for binary concepts and to the empirical mean for continuous ones. Attribution patterns reveal the extent to which the model relies on concepts vs. residuals, and how this balance shifts under decorrelation.

## 6 Experiments

In this section, we evaluate CRMs by exploring the following questions:

• **Q1: Concept-Residual Overlap**: Does concept-residual overlap exist in CRMs, and how does it change with increasing residual capacity? Can MI-based regularization suppress this overlap more effectively than existing decorrelation methods?

• **Q2: Concept Importance**: As residual dimension—and thus concept-residual overlap—increases, how does this affect the functional importance of concept representations in CRMs? Does MI-based disentanglement better preserve reliance on the concept layer compared to alternative methods?

• **Q3a: Impact on Baseline Performance**: How does MI minimization's constraint on concept-residual overlap affect task performance compared to other CRM disentanglement techniques and CEMs?

• **Q3b: Intervention Efficacy**: Does MI minimization's reduction of concept-residual overlap enhance CRMs' responsiveness to test-time concept interventions compared to alternative approaches?

**Datasets and Tasks**  We analyze five vision tasks: (1) **CUB**, a bird classification task with the 112-concept subset (of 312 original attributes) from Koh et al. (2020) predicting 200 species (Wah et al., 2011), (2) **OAI**, using ten ordinal variables to predict four KL grades of knee osteoarthritis severity (Nevitt et al., 2006), (3) **CIFAR-100**, with 20 superclasses as concepts where each image belongs to one of five associated classes per superclass (Krizhevsky et al., 2009), (4) **CelebA**, using 6 of 8 concept annotations with vector product creating a 256-class task (Espinosa Zarlenga et al., 2022; 2023), and (5) **AA2**, where we use a 6-concept subset to create an incomplete task allowing only 38 of 50 classes to be uniquely distinguished (Xian et al., 2019). Complete datasets (CUB, OAI) contain sufficient concepts to predict the task, while incomplete ones (CIFAR-100, CelebA, AA2) require additional information for high task performance.

**Models**  We evaluate mutual information based Concept Residual Models against other disentanglement techniques including decorrelation loss, iterative normalization (Huang et al., 2019), and expert yielded estimates (EYE) regularization (Wang et al., 2022). We compare these approaches with baseline CBMs and CEMs using identical architectures, averaging metrics over five random initializations across challenging image classification and medical datasets. To match previous papers, we use ResNet18 pre-trained on ImageNet as the backbone for CIFAR, AA2, and OAI; ResNet34 for CelebA; and InceptionV3 for CUB. The output of the feature extractor is split into concept and residual layers. The target network consists of one layer for CIFAR, CelebA, and CUB, and three layers for OAI. Semi-independent training is used for all datasets except for CelebA, where disentangled intervention aware training is used for all D-CRMs. CEMS are trained using intervention aware training with hyper parameters supplied in Espinosa Zarlenga et al. (2022). Further details on Disentangled Intervention Aware Training can be found in Section D.2 and in our code included in the supplementary material.

| | Incomplete Concept Sets | | | | | | | | Complete Concept Sets | | | | | | | |
| | CIFAR 100 | | | | CelebA | | | | CUB | | | | OAI | | | |
| Method | F1↓ | F1$_\rho$↓ | MI↓ | MI$_\rho$↓ | F1↓ | F1$_\rho$↓ | MI↓ | MI$_\rho$↓ | F1↓ | F1$_\rho$↓ | MI↓ | MI$_\rho$↓ | RMSE↑ | RMSE$_\rho$↓ | MI↓ | MI$_\rho$↓ |
|---|---|---|---|---|---|---|---|---|---|---|---|---|---|---|---|---|
| Latent | 0.62 | 1.00 | 1.51 | 1.00 | 0.87 | 0.75 | 2.69 | 0.99 | 0.29 | 1.00 | 1.26 | 0.89 | 0.99 | 0.96 | 9.49 | 0.93 |
| Decorr. | 0.61 | 0.98 | 1.45 | 1.00 | 0.87 | 0.62 | 2.65 | 0.88 | 0.26 | 1.00 | 0.78 | 0.89 | 1.18 | 0.86 | 6.54 | 0.86 |
| IterNorm | 0.62 | 0.91 | 1.39 | 1.00 | 0.80 | 0.98 | 2.22 | 0.98 | 0.25 | 0.89 | 0.77 | 1.00 | 1.00 | 0.82 | 8.54 | 0.96 |
| EYE | 0.61 | 0.98 | 1.17 | 1.00 | 0.84 | 0.95 | 2.39 | 0.93 | 0.28 | 0.93 | 0.77 | 0.93 | 0.96 | 0.95 | 9.15 | 0.93 |
| Adv. Decorr. | 0.33 | *0.64* | 0.87 | *0.75* | 0.84 | 0.97 | 2.23 | 0.97 | 0.22 | 0.96 | 0.75 | *0.86* | 0.85 | 0.96 | 8.83 | 1.00 |
| MI | **0.12** | *0.50* | **0.70** | *0.19* | **0.78** | 0.99 | **1.50** | 0.99 | **0.22** | 0.75 | **0.70** | 0.82 | **1.34** | *0.25* | **1.25** | *0.39* |
| Random | 0.05 | - | - | - | 0.471 | - | - | - | 0.15 | - | - | - | 1.30 | - | - | - |

Table 1: Summary of concept-residual overlap across disentanglement strategies. F1-Score and Mutual Information (MI) are reported at a fixed high residual dimensionality (16 for all datasets, 64 for CUB), with lower values indicating better disentanglement. To assess monotonic trends, we compute Spearman's rank correlation coefficient ($\rho$) between residual dimensionality (powers of 2 from 1 to 64) and the corresponding F1/MI values; these are shown in subscripted columns. **Bold values** denote statistically significant improvements ($p < 0.01$) in the primary metric. Italicized $\rho$ values indicate no evidence of a significant monotonic relationship ($p \geq 0.01$). Random baselines are reported for F1 only.

# 7  Results

Across five datasets encompassing both complete and incomplete concept scenarios, we demonstrate that standard CRMs exhibit significant concept-residual overlap (F1-scores: 0.29-0.87, MI scores: 1.17-2.69).

Our proposed MI-based D-CRMs reduce this overlap by 61-81% while maintaining performance within 3% of baseline CRMs. This overlap reduction yields substantial improvements in intervention effectiveness (up to 16% on OAI) and concept importance preservation (3-57x), validating our core hypothesis.

## 7.1 Concept-Residual Overlap in Standard CRMs (RQ1)

Our first research question asks whether concept-residual overlap exists in CRMs and how effectively MI-based regularization can suppress it. Our hypothesis is that as the residual layer's capacity increases, it encodes more concept-related information, undermining model interpretability. We find that standard CRMs exhibit significant overlap that increases monotonically with residual capacity, while MI-based D-CRMs achieve near-complete suppression on 2/4 tested datasets and substantial reduction on all others, outperforming all alternative disentanglement methods.

We systematically evaluate concept-residual overlap by training CRMs with residual dimensions ranging from 1 to 64 (powers of 2) and measuring both F1-Score (ability to predict concepts from residuals) and Mutual Information between concepts and residuals. To validate monotonic relationships, we compute Spearman's rank correlation coefficients ($\rho$) between residual dimension and overlap metrics. Table 1 summarizes our findings at high residual capacities, with correlation coefficients quantifying the strength of monotonic trends.

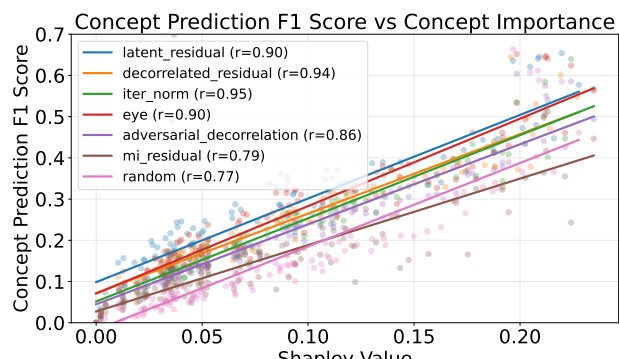

Figure 3: F1-Scores for each concept plotted against the concepts' estimated Shapley value for a residual dimension of 64 for the CUB dataset. Results show the residual encodes more information on concepts that are found to be important to the task prediction.

**Standard CRMs exhibit strong concept-residual overlap.** Latent residual CRMs show strong positive correlations between residual size and overlap across all datasets: F1-Score correlations of $\rho = 1.00$ (CIFAR-100, CUB), $\rho = 0.75$ (CelebA), and $\rho = 0.96$ (OAI). Mutual information correlations are similarly high: $\rho = 1.00$ (CIFAR-100), $\rho = 0.99$ (CelebA), $\rho = 0.89$ (CUB), and $\rho = 0.93$ (OAI). These results provide strong evidence that residual capacity directly enables concept information leakage in standard CRMs.

**Residuals selectively encode task-relevant concepts.** Figure 3 reveals that standard CRMs exhibit strong correlations ($R^2 > 0.80$) between concept importance (SHAP values) and residual encoding strength (F1-scores), significantly exceeding random baseline correlations ($p < 0.01$). This demonstrates that overlap is not random but systematically biased toward task-critical concepts, making the interpretability problem even more severe.

**MI-based D-CRMs achieve superior overlap reduction.** As shown in Table 1, MI D-CRMs consistently achieve the lowest overlap across all datasets and metrics. The improvements are substantial: F1-score reductions of 81% (CIFAR-100), 10% (CelebA), 24% (CUB) relative to the next-best method, and RMSE improvements of 34% (OAI). All improvements are statistically significant ($p < 0.01$). Importantly, MI D-CRMs reduce the correlation between concept importance and residual encoding to $R^2 = 0.63$, which is not significantly different from the random baseline ($R^2 = 0.59$), indicating more uniform disentanglement.

**Near-complete suppression achieved on CIFAR-100 and OAI.** On these datasets, MI D-CRMs break the monotonic relationship between residual size and overlap: Spearman correlations become non-significant (CIFAR-100: $\rho = 0.50, p = 0.21$; OAI: $\rho = 0.25, p = 0.58$). Furthermore, F1-scores of 0.12 (CIFAR-100) and RMSE of 1.34 (OAI) are statistically indistinguishable from random baselines of 0.05 and 1.30 respectively ($p < 0.01$), indicating near-complete disentanglement.

**Partial but significant reduction on CelebA and CUB.** While MI D-CRMs do not fully eliminate overlap on these datasets (F1-scores of 0.78 and 0.22 remain above random baselines), they still achieve the

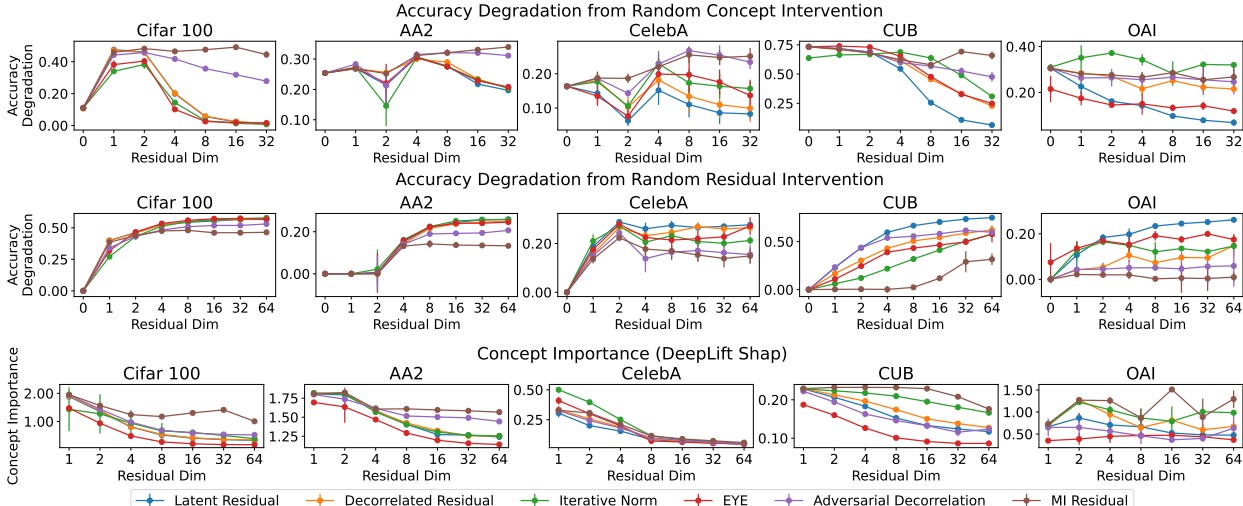

Figure 4: Test-time intervention results averaged over 5 seeds. **Top**: Random concept interventions; higher degradation indicates stronger concept dependence. **Middle**: Random residual interventions; lower degradation suggests reduced residual reliance. **Bottom**: DeepLIFT SHAP concept importance; higher values indicate greater concept influence.

best performance among all tested methods, with statistically significant improvements in both F1-score and MI metrics. CelebA presents a particularly challenging case where all methods, including MI, show uniform decline patterns, though MI consistently outperforms alternatives across all residual dimensions.

## 7.2 Impact of Overlap on Concept Importance (RQ2)

Having established that concept-residual overlap exists, we next investigate how this overlap affects the functional importance of concepts in CRM decision-making. Our analysis reveals that overlap progressively diminishes concept importance as residual capacity increases, with models increasingly relying on opaque residual pathways. However, MI-based D-CRMs preserve concept importance three times better than standard CRMs, maintaining interpretable decision-making even at high residual dimensions.

We assess concept importance through three complementary metrics: random concept interventions (measuring performance degradation when concepts are corrupted), SHAP-based attribution scores, and random residual interventions (measuring residual dependence).

**MI D-CRMs maintain significantly stronger concept reliance.** Under random concept interventions (Figure 4, top), MI D-CRMs exhibit substantially larger accuracy degradations than competing methods, indicating preserved concept dependence. At residual dimension 64: CIFAR-100 shows 40.1% degradation (vs. 0.7% for latent residual), CUB shows 57.4% degradation (vs. 4.4%), and CelebA shows 22.0% degradation (vs. 6.7%). This 6–57× difference demonstrates that MI D-CRMs successfully prevent the model from bypassing the concept layer.

**SHAP-based concept attribution reveals preserved reliance.** Figure 4 (bottom) shows that MI D-CRMs maintain higher concept attribution scores across increasing residual dimensions. On CIFAR-100, MI preserves a SHAP score of 1.01 at dimension 64—over 3× higher than latent residual (0.33) and decorrelated residual (0.34). This pattern holds across CUB (0.18 vs 0.12) and CelebA datasets. ANOVA confirms statistical significance ($p < 6 \times 10^{-5}$) across all residual dimensions >4.

**Random residual interventions reveal reduced dependence on residual features.** Random residual interventions (Figure 4, middle) reveal that MI D-CRMs maintain lower dependence on residual features. On complete concept datasets where residuals should be unnecessary: OAI shows <1% accuracy reduction for MI vs. 26% for latent residual; CUB shows 31.6% degradation for MI vs. 74.9% for latent residual. On incomplete concept datasets where some residual dependence is expected, MI still shows reduced reliance: CIFAR-100 (38.6% vs. 56.6%), CelebA (17.1% vs. 27.6%).

| Method | Incomplete Concept Sets | | | | | | Complete Concept Sets | | | |
| | CIFAR 100 | | AA2 | | CelebA | | CUB | | OAI | |
| | B | $C^+\uparrow$ | B | $C^+\uparrow$ | B | $C^+\uparrow$ | B | $C^+\uparrow$ | B | $C^+\uparrow$ |
|---|---|---|---|---|---|---|---|---|---|---|
| Bottleneck | 0.11 | 0.2 | 0.28 | 0.79 | 0.24 | 0.27 | 0.75 | 0.99 | 0.69 | 0.96 |
| Latent | **0.60** | 0.66 | **0.42** | 0.90 | 0.35 | 0.70 | **0.79** | 0.93 | **0.72** | 0.78 |
| Decorrelation. | **0.60** | 0.67 | **0.42** | 0.90 | 0.35 | 0.70 | **0.79** | 0.94 | **0.72** | 0.82 |
| Iterative Normalization | **0.60** | 0.67 | **0.42** | 0.90 | 0.35 | 0.71 | **0.79** | 0.96 | 0.71 | 0.90 |
| EYE | 0.59 | 0.61 | 0.42 | 0.90 | **0.37** | 0.67 | 0.77 | 0.95 | 0.72 | 0.82 |
| Concept Embedding Model | 0.44 | 0.70 | 0.33 | 0.89 | 0.24 | 0.67 | 0.62 | **0.99** | - | - |
| Adversarial Decorr. | 0.56 | 0.68 | 0.40 | **0.91** | 0.36 | 0.72 | 0.74 | 0.89 | 0.65 | 0.89 |
| Mutual Information | 0.58 | **0.72** | 0.40 | **0.91** | 0.35 | **0.73** | 0.76 | 0.96 | 0.67 | **0.94** |

Table 2: Classification accuracy comparison: baseline (B) and positive concept interventions ($C^+$). Bold values indicate statistically significant best performance ($p < 0.05$). MI consistently achieves superior intervention performance despite modest baseline costs.

**Disentanglement Preserves Counterfactual Reasoning.** Using the AA2 dataset where 'white' is the only distinguishing feature between brown and polar bears, we test counterfactual reasoning by intervening on brown bear images to set 'white'=true. Figure 5 shows that while CBMs achieve perfect 100% counterfactual accuracy, standard CRMs fail increasingly as residual dimension grows (dropping to ~25% at dimension 64). MI D-CRMs maintain robust counterfactual performance (98% accuracy), demonstrating that disentanglement preserves concept-level reasoning capabilities.

### 7.3 Performance Trade-offs and Intervention Efficacy (RQ3a & RQ3b)

Our final research questions examine whether MI-based disentanglement affects baseline task performance and intervention effectiveness. We find that while MI-based disentanglement incurs modest baseline accuracy drop (1-3%), it substantially improves intervention efficacy, achieving the highest positive concept intervention accuracy across 4/5 datasets with improvements up to 16%. These results validate that the practical benefits of reduced concept-residual overlap outweigh the minimal performance trade-offs.

**Acceptable performance trade-offs for interpretability gains.** Table 2 shows that MI D-CRMs achieve baseline accuracies within 1-5% of standard CRMs across all datasets. Specifically: CIFAR-100 (0.58 vs 0.60), AA2 (0.40 vs 0.42), CUB (0.76 vs 0.79), and OAI (0.67 vs 0.72). Only CelebA maintains equivalent performance (0.35). These modest costs are consistent with other interpretability methods and represent acceptable trade-offs given the substantial interpretability benefits.

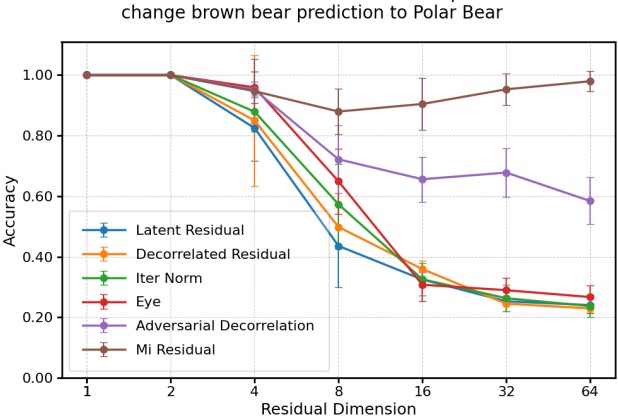

Figure 5: Counterfactual intervention accuracy on the 'white' concept for brown bear images in the AA2 dataset, testing whether models update predictions to polar bear when the distinguishing concept is modified.

**Superior intervention efficacy validates practical utility.** MI D-CRMs achieve the highest positive concept intervention accuracy on 4/5 datasets, with substantial improvements: CIFAR-100 (0.72 vs 0.66, +6%), AA2 (0.91 vs 0.90, +1%), CelebA (0.73 vs 0.70, +3%), and OAI (0.94 vs 0.78, +16%). These improvements demonstrate that reduced concept-residual overlap directly translates to enhanced human-model interaction capabilities, validating the core premise that disentanglement improves practical interpretability.

**Comparison with Concept Embedding Models.** CEMs, which use more complex training procedures, show larger baseline performance drops (0.44 vs 0.58 on CIFAR-100) while achieving comparable intervention performance (0.70 vs 0.72). This suggests MI D-CRMs provide a superior balance between baseline performance and intervention effectiveness.

## 8 Conclusion

Our study systematically demonstrates that standard Concept-Residual Models (CRMs) suffer from *concept-residual overlap*, a critical flaw where the residual pathway redundantly encodes concept information. We show not only that this overlap increases monotonically with residual capacity, but that it is systematically biased, with the residual preferentially encoding concepts most critical to the downstream task (Figure 3). This phenomenon fundamentally undermines the core goals of concept-based models: interpretability and reliable human-in-the-loop intervention.

To address this, we introduced Disentangled Concept-Residual Models (D-CRMs) and a novel mutual information (MI) minimization objective. Our experiments across five diverse datasets validate this approach, showing that MI-based D-CRMs reduce concept-residual overlap by up to 81% and achieve near-complete disentanglement on challenging datasets like CIFAR-100 and OAI. This disentanglement restores the model's functional reliance on the interpretable concept pathway. As shown in Section 7.2, MI D-CRMs preserve concept importance up to 57× better than standard CRMs under random concept interventions and maintain 98% accuracy in counterfactual reasoning tasks where standard CRMs fail (dropping to 25% accuracy).

Crucially, these gains in interpretability and robustness do not come at a significant cost to performance. Our MI-based D-CRMs maintain baseline accuracy within 3% of standard CRMs while substantially improving intervention efficacy by up to 16% (Table 2). By successfully balancing predictive performance with interpretability, MI-based D-CRMs represent a significant step forward. They are a practical and effective solution for building more reliable and transparent concept-based models, broadening their applicability in high-stakes domains where both accuracy and human oversight are paramount.

**Limitations and Future Work** While promising, our approach has limitations that suggest clear directions for future work. The MI minimization does not completely eliminate overlap on all datasets, and it introduces a modest performance trade-off. Furthermore, the scalability of our method to massive, ImageNet-scale datasets remains an open question. Future work could explore more powerful disentanglement techniques to address the remaining overlap and develop methods to close the performance gap. Finally, creating large-scale concept benchmarks would be a significant contribution to the field, allowing for more rigorous testing of the next generation of interpretable models.

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

## A    Additional Background

### A.1    Concept Embedding Models

Concept Embedding Models (CEMs) relax the strict bottleneck constraint imposed by CBMs by learning high-dimensional, continuous embeddings for each concept, rather than forcing predictions to rely solely on discrete, human-interpretable concept labels. Specifically, Concept Embedding Models (CEMs) learn high-dimensional concept embeddings through a concept encoder $g : \mathbb{R}^d \to \mathbb{R}^{k \cdot m}$ that maps input $X$ to concept embeddings, where $m$ is the embedding dimension for each of the $k$ concepts (Espinosa Zarlenga et al., 2022; 2023). For each concept $c_i$, CEMs learn two semantic embeddings: $\hat{c}_i^+$ and $\hat{c}_i^-$ representing the positive and negative states, respectively. The final concept embedding $\hat{c}_i$ is computed as a weighted mixture:

$$\hat{c}_i = \hat{p}_i \hat{c}_i^+ + (1 - \hat{p}_i)\hat{c}_i^- \tag{15}$$

where $\hat{p}_i = \sigma(W_s[\hat{c}_i^+, \hat{c}_i^-]^T + b_s)$ is the predicted probability that the concept is active, calculated using a shared scoring function $s : \mathbb{R}^{2m} \to [0, 1]$. The downstream model $f : \mathbb{R}^{k \cdot m} \to \mathbb{R}$ maps these concept embeddings to the target prediction.

By embedding concepts in a continuous latent space and combining them in a soft, learnable manner, CEMs provide greater representational flexibility than CBMs. This allows them to capture subtle or complex variations in concept instantiations that may be lost in a discrete bottleneck. As a result, CEMs effectively relax the bottleneck assumption by enabling the model to represent and use more nuanced information for prediction, even when the concept set is incomplete or ambiguous. However, this added flexibility comes at the cost of interpretability and counterfactual reasoning. Since each embedding combines concept activation with implicit residual structure, modifying a concept affects both its representation and its interactions with the rest of the embedding space. This entanglement makes it difficult to isolate the causal impact of individual concepts or to reason about how concept changes alone influence model predictions.

## B    Dataset Details

**CIFAR-100** The CIFAR-100 dataset is uniquely utilized by employing its 20 superclasses as primary concepts (Krizhevsky et al., 2009). Given a superclass, an image can belong to any of the five associated classes, indicating that knowledge of the superclass alone is insufficient for precise classification, as the concept space is incomplete. The challenge lies in predicting one of 100 classes, which requires the residual to encode the conditional distribution of possible classes given a superclass. This setup makes CIFAR-100 an ideal benchmark for evaluating methods that can disentangle the residual and concept layers while preserving valuable information in both.

**CelebA** For CelebA, we follow Espinosa Zarlenga et al. (2022), selecting the 8 most balanced attributes [a1, · · · a8] out of each image's 40 binary attributes, as defined by how close their distributions are to a random uniform binary distribution, and use attributes [a1, · · · , a6] as concepts annotations for each sample. To create a scenario where full concept annotations are not available, each image in CelebA receives a label that matches the decimal value of the binary vector [a1, ..., a8], leading to a total of $l = 2^8 = 256$ classes. It is important to note that the concept annotations are partial, as attributes a7 and a8, crucial for the downstream task, are omitted from the concept set. To enhance training efficiency and resource use, the size of the CelebA dataset is decreased by selecting every 12th sample through random subsampling and reducing each image's resolution to (3, 64, 64). Consequently, this process yields approximately 16,900 RGB images, which are then divided into training, validation, and test sets following a conventional 70%-10%-20% distribution.

**Caltech-UCSD Birds (CUB)**: The CUB dataset is a rich resource for fine-grained visual categorization, particularly in bird species recognition (Welinder et al., 2010). It contains images of 200 bird species, each annotated with detailed attributes like feather color, beak shape, and wing patterns. These concepts are learned via multi-binary classification (Chen et al., 2020). Bottleneck Models trained on CUB rely heavily on the use of majority voting (Chen et al., 2020). Chen et al. explain that the provided concepts are noisy,

and thus to make them effective for CBMs they employ majority voting; if more than 50% of a downstream class has a particular concept in the data, then we set all that downstream task to have that concept.

**OAI (Osteoarthritis Initiative)**: The OAI dataset, comprising 36,369 knee X-ray data points, focuses on individuals at risk of knee osteoarthritis. It includes both radiological and clinical data, with the primary task being to predict the Kellgren-Lawrence grade (KLG), a four-level ordinal scale used by radiologists to assess osteoarthritis severity. Higher KLG scores indicate more severe disease. The dataset utilizes ten ordinal variables as concepts for analysis, encompassing joint space narrowing, bone spurs, calcification, and other clinical features. These variables, essential for evaluating osteoarthritis severity, align with the preprocessing techniques employed by Pierson et al. (2021).

## C   Experiment Details

### C.1   Model Architecture

We used pretrained imagenet models as our backbones. For CUB, we use inception_v3 (Szegedy et al., 2016), following Chen et al. (2020). For CIFAR and OAI, we use resnet18 (He et al., 2016). For CelebA, we follow Espinosa Zarlenga et al. (2022) and use resnet34. The target network is one layer for CIFAR, CelebA and CUB, and three for OAI, with each hidden layer being 50 neurons. Specific hyperparameters and detailed configurations for each dataset can be found in the configs/ directory of the provided codebase.

### C.2   Training Scheme

We apply intervention-aware training only to the **CelebA** dataset, where we observe consistent improvements in both baseline accuracy and intervention performance (typically 1-2%) compared to standard independent training. For **CUB**, **OAI**, and **CIFAR-100**, we do not use intervention-aware training, as empirical evaluation shows no statistically significant benefit over semi-independent training.

**Concept Embedding Models (CEMs)**   We employ intervention-aware training and maintain the original hyperparameters as specified in the source implementation. However, a key distinction in our approach is the omission of concept groups. Instead of using hundreds of concept embeddings for the CUB dataset as in the original implementation, we reduce this to 28 core concepts. This simplification likely explains the lower baseline performance we observe on CUB compared to the results reported in the original paper. Specific hyperparameters and detailed configurations for each dataset can be found in the configs/ directory of the provided codebase.

## D   Additional Methodology Details

### D.1   EYE Regularization

Concept-Credible models propose **EYE regularization**, which targets the dependencies on features correlated with but not within concept space $C$ (Wang et al., 2022). Let $\theta_x$ and $\theta_c$ be the parameters for the linear transformation from the concept and residual layers. The combined penalty function, $J([\theta_x, \theta_c]) = |\theta_x|_1 + \sqrt{|\theta_x|_2^2 + |\theta_c|_2^2}$. The EYE penalty imposes a stricter penalty on $\theta_x$ relative to $\theta_c$, permitting a greater norm for $\theta_c$ and thus promoting reliance on the concept space $C$.

While EYE regularization does not explicitly constrain the residual representations themselves, it creates an implicit regularization effect through backpropagation. By differentially penalizing how the residual layer's outputs are transformed in the downstream network ($\theta_c$), the gradients flowing back to the residual encoder will favor learning representations that are complementary to, rather than redundant with, the concept space. This occurs because the network optimizes to make efficient use of the less-penalized concept pathway, incentivizing the residual encoder to capture only the information that cannot be effectively encoded through concepts. However, this implicit approach to disentanglement may not be as robust as methods that directly constrain the statistical independence between concept and residual representations, such as mutual information minimization.

### D.2 Disentangled Intervention aware Training

For CelebA, we extend our semi-independent training objective with disentangled intervention-aware training, following the framework introduced in Intervention-Aware Concept Embedding Models (IntCEM) (Espinosa Zarlenga et al., 2023). As in the semi-independent setting, the model comprises: (1) a concept encoder $g : \mathcal{X} \to \hat{c}$, (2) a residual encoder $h : \mathcal{X} \to r$, and (3) a predictor $f : \mathcal{C} \times \mathcal{R} \to \hat{y}$, where $x \in \mathcal{X}$, $r = h(x)$, and the concept input to the predictor is an intervened version $\tilde{c}$ of $g(x)$.

**Intervention-based supervision.** During training, we sample a stochastic initial intervention mask $\mu^{(0)} \sim \text{Bernoulli}(p_{\text{int}})$ and a trajectory length $T \sim \text{Unif}(\{1, \ldots, T_{\max}\})$. A sequence of interventions is applied to the predicted concepts $\hat{c} = g(x)$, replacing selected dimensions with their corresponding ground truth values $c$, producing an intervened vector $\tilde{c}(x, c, \mu^{(0)}, T)$. This process partially supervises the concept space while encouraging robustness to interventions.

**Training losses.** The final training objective becomes:

$$\min_{g,h,f} \mathbb{E}_{(x,c,y)\sim\mathcal{D}} \mathbb{E}_{\substack{\mu^{(0)}\sim\text{Bern}(p_{\text{int}}) \\ T\sim\text{Unif}(\{1,\ldots,T_{\max}\})}} \left[ \alpha \underbrace{\mathcal{L}_{\text{concept}}(g(x), c)}_{\text{Concept supervision}} + \beta \underbrace{\mathcal{L}_{\text{target}}(f(\tilde{c}(x, c, \mu^{(0)}, T), h(x)), y)}_{\text{Trajectory-aware prediction}} \right.$$
$$\left. + \gamma \underbrace{\mathcal{L}_{\text{disentangle}}(h(x), c)}_{\text{Disentanglement}} \right] \tag{16}$$

where $\tilde{c}(x, c, \mu^{(0)}, T)$ denotes a $T$-step intervention trajectory that replaces progressively more components of $g(x)$ with corresponding values from $c$.

In practice, we sample a single $\mu^{(0)}$ and $T$ per training step, with $p_{\text{int}} = 0.25$ and $T_{\max}$ linearly annealed from 2 to 6 during early training.

## E   Further Discussion

### E.1 CEM Baseline Performance

As discussed in Section 7.3, CEMs exhibit consistently lower baseline performance across all of our tested datasets. This reflects a fundamental trade-off in CEMs, which incorporate an intervention-aware mechanism that can be tuned to prioritize either baseline or intervention performance. Their behavior is governed by several hyperparameters (e.g., $\lambda_{\text{roll}}$, $\lambda_{\text{concept}}$), allowing practitioners to emphasize one objective over the other. In our experiments, we adopt the parameters reported in prior work that maximize intervention performance for fair comparison to our work, aligning with the goal of this paper to evaluate and improve model intervenability.

