# OpenReview forum: "Disentangled Concept-Residual Models: Bridging the Interpretability–Performance Gap for Incomplete Concept Sets"
_TMLR — Accepted by TMLR_

### Review · Reviewer_vKDv · 2025-09-30

**Summary Of Contributions:**

This paper proposes Disentangled Concept-Residual Models to mitigate the problem of concept–residual overlap in Concept Residual Models. The authors formally define and measure this overlap, introduce new interpretability metrics, and present a mutual information minimization approach that reduces overlap while preserving accuracy. Experiments on multiple datasets (CUB, CIFAR-100, CelebA, AA2, OAI) demonstrate incremental interpretability, robustness of interventions, and strong predictive performance. Strengths include clearly demonstrated motivation, a well-structured paper organization, and comprehensive experiments. Weaknesses lie in limited baseline comparisons, lack of human evaluation, and some minor presentation issues.

**Additional Comments:**

N/A

**Audience:**

Yes

**Audience Explanation:**

The paper introduces D-CRMs as a practical solution to concept–residual overlap, a major limitation of Concept Bottleneck and Residual Models. This contribution is especially relevant for the XAI community as it improves interpretability, supports more reliable human interventions, and narrows the gap between performance and transparency. The method is rigorous, combining a mutual information minimization objective with newly designed interpretability metrics.

**Broader Impact Concerns:**

No broader impact

**Claims And Evidence:**

No

**Claims Explanation:**

- Novel interpretability metrics. The paper introduces random concept and random residual intervention experiments. These are clearly defined and implemented in the evaluation section, effectively highlighting differences in prediction reliance between standard CRMs and the proposed D-CRMs

- Mutual information minimization objective. The disentanglement objective (Eq. 6–8) is formally defined and validated in ablation studies. Results show a 61–81% reduction in concept–residual overlap across datasets

- Comprehensive empirical evaluation. The model is tested on five diverse datasets (CUB, CIFAR-100, CelebA, AA2, OAI), covering both complete and incomplete concept sets. Experimental results demonstrate that D-CRMs restore interpretability and intervention reliability.

- Tradeoff between performance and interpretability. The paper carefully quantifies baseline accuracy costs (typically ≤3% compared to CRMs), demonstrating that the modest reduction in raw accuracy is outweighed by substantial gains in interpretability and human-intervention reliability

**Requested Changes:**

- More baseline comparisons. The experimental validation does not include comparisons against more recent methods that address interpretability or disentanglement. For example:

  - Concept Realignment for CBMs (Nishad et al., ECCV 2024) use post-intervention to realign correlated concepts to boost intervention efficacy.

  - Vision-Language Guided CBMs (Srivastava et al., NeurIPS 2024) that leverages multimodal signals to improve interpretability.


- Scale on larger dataset. All experiments are conducted on small- to medium-scale datasets (CUB, CIFAR-100, CelebA, AA2, OAI). It remains unclear whether D-CRMs scale effectively to larger, more complex datasets. The authors should provide an additional experiment on a larger benchmark (e.g., ImageNet subset with defined attributes).

- Human Evaluation. Although the paper proposes new interpretability metrics, it does not include any human evaluation to confirm whether the disentangled concepts align with human understanding. This omission limits the practical relevance of the work for explainable AI. The authors should consider adding a small-scale user study (e.g., a survey asking participants to judge concept clarity and usefulness).

- Minor formatting/presentation issue.

  - inconsistent dash vs. hyphen, e.g., "concept–residual overlap" and "concept-residual overlap".

  - Fig.4, overlapped vertical label (row 2 & 3).

  - The term baseline performance costs (Section 7.3) might be clearer as baseline accuracy drop or performance trade-off.

  - Yuksekgonul et al. is cited twice (conference and arXiv preprint).

  - Some venues are abbreviated (ICML, ECCV) while others are written in full (European Conference on Computer Vision).

---

### Review · Reviewer_YoDQ · 2025-10-01

**Summary Of Contributions:**

This paper addresses the problem of concept–residual overlap in Concept-Residual Models (CRMs), where residual pathways can encode redundant concept information and undermine interpretability. The authors propose to measure overlap via probes and mutual information, evaluate reliance with intervention-based metrics, and reduce overlap using cross-covariance decorrelation and a CLUB-based MI regularizer. Experiments across multiple datasets suggest that residuals indeed contain concept information, and that their methods reduce overlap while preserving predictive accuracy.

**Strengths:**
- Clearly motivates why residuals can undermine CBM interpretability
- Proposes two disentanglement objectives that are technically simple and empirically effective
- Evaluates on multiple datasets with residual dimension sweeps

**Weaknesses:**
- The novelty of the identified problem is overstated. The paper often reads as if the authors are the first to identify this problem, but it was already discussed in prior work, e.g. [1] introduces an orthogonality loss for this purpose.
- The “formal definition” of overlap (see contributions) is not formal in the mathematical sense; it is operationalized via probes and MI bounds.
- Since it partly operationalises concept overlap via probes, it misses adversarial decorrelation as a baseline.
- The interventions are interesting but seem to have limitations in practice: If predicted concepts are already close to ground-truth, ACC_pos adds almost nothing. Small gains could be misinterpreted as “label ignores concepts” when in fact nothing changed. Similarly, some concepts may not affect the label at all. Intervening on them yields no effect, but this does not prove residual reliance.

[1] A. A. Ismail, J. Adebayo, H. C. Bravo, S. Ra, and K. Cho, ‘Concept Bottleneck Generative Models’, in The Twelfth International Conference on Learning Representations, 2024.

**Audience:**

Yes

**Audience Explanation:**

Yes, researchers working on interpretability, concept bottleneck models, and disentangled representations would likely be interested in the paper’s findings on residual-concept overlap and methods to mitigate it.

**Broader Impact Concerns:**

No.

**Claims And Evidence:**

Yes

**Claims Explanation:**

The authors demonstrate that residual representations contain information correlated with concepts and provide evidence that MI-based regularizers can reduce this correlation and, in some settings, improve steerability. This constitutes meaningful empirical support for their central claim that residual channels undermine interpretability. At the same time, the scope of what is demonstrated is narrower than how some claims are phrased. The definition of overlap relies on probe accuracy and CLUB-based mutual information estimates, which establish statistical dependence but not necessarily semantic or causal redundancy. Similarly, the intervention results show that modifying concepts or residuals can affect predictions, but because interventions are applied to random concepts, they sometimes operate on already-correct or label-irrelevant dimensions, making the observed effects difficult to interpret as evidence that the model is systematically relying on residuals over concepts.

**Requested Changes:**

Critical:
- Acknowledge that residual–concept overlap is a recognized problem, with prior work already trying to address it (e.g., orthogonality loss)
- The current operationalization via probes and MI estimates is useful but does not constitute a formal definition of overlap; the paper should avoid overstating this
- Respond to concerns about ambiguity of interventions (see above)

Optional:
- Compare against adversarial decorrelation as a baseline

---

### Review · Reviewer_6jG2 · 2025-10-20

**Summary Of Contributions:**

The paper proposes a variant of concept-residual models (CRM) that reduces “concept-residual overlap” that they call disentangled-CRM (D-CRM). This attempts to minimize an estimate of the mutual information between concept and residual via a learned variational approximation that is jointly trained with the main model. Experiments show that this removes concept-residual overlap while maintaining performance.

**Audience:**

Yes

**Audience Explanation:**

Yes, this seems to be a clear improvement over CRMs in terms of interpretability by preventing the model from pushing all information into the residuals.

**Claims And Evidence:**

Yes

**Claims Explanation:**

Yes, the paper provides a good summary of related work, clear explanation of the proposed method, and sufficient experimentation on a few datasets that clearly illustrates how concept-residual overlap is an issue with CRMs but not D-CRMs with the proposed mutual information objective.

**Requested Changes:**

Minor: the DeepLift reference is broken in section 5.2.

---

> ### Author Response · Authors · 2025-10-29
> **Response to Reviewer 6jG2**
>
> We thank Reviewer 6jG2 for the thoughtful and positive assessment of our framing, method, and experiments. We are encouraged by the reviewer’s conclusion that our approach reduces concept–residual overlap while maintaining performance and improves interpretability by preventing information from being pushed into residuals.
>
> Action taken:
> - We fixed the DeepLIFT reference and standardized the method name to “DeepLIFT” throughout. We verified that the citation compiles and links correctly.

---

### Decision · Action_Editor_8rsc · 2025-12-23

**Recommendation:** Accept as is

**Audience:**

Yes

**Audience Explanation:**

I believe researchers working on explainable machine learning and, in particular, on concept-based explanations, will consider this work relevant to the field, as it proposes well-motivated, simple and effective metrics and mitigation measures for concept-residual overlap.

**Claims And Evidence:**

Yes

**Claims Explanation:**

Reviewers acknowledged the thorough motivation for the proposed metrics and mitigation measures, as well as the broad set of considered datasets.
However, several concerns were raised as well.
For instance, it was brought up that the novelty of the concept-residual overlap problem is overstated, that prior work has already attempted to address the problem using decorrelation (adversarial decorrelation), and  that some representative baselines were missing (including but not limited to adversarial decorrelation). Further, it was pointed out that the proposed intervention-based metrics were not clearly described and potentially ambiguous.

The authors' rebuttal addressed these points by:
* adding adversarial correlation as a baseline and discussing why other approaches are not directly comparable (different intervention protocol) or provide orthogonal benefits (automated discovery of concepts, while this work assumes available ground-truth concepts)
* toning down the novelty claims, and
* providing the very important clarification that all concepts are intervened on **simultaneously**, which addresses concerns about ambiguity raised by Reviewer YoDQ.

I recommend acceptance of the manuscript, provided the requested changes listed below are applied for the camera-ready submission. I believe that the main concerns of reviewers have been appropriately addressed. The clarification regarding the intervention setup, in particular, convinced me that the claims are supported by accurate, convincing, and clear evidence.

## Required changes for camera-ready version:
* citation formatting: please use \citep for parenthetical references, e.g., (Huang et al., 2019), when the work or authors are not part of the sentence. Currently, the manuscript seems to use \citet (or another command that produces equivalent references) everywhere. See the below example taken from page 8 (in the "Models" paragraph).
The following sentence

    > [...]techniques including decorrelation loss, iterative normalization **Huang et al. (2019)**, and expert yielded estimates (EYE) regularization **Wang et al. (2022)**.

    should be
    > [...]techniques including decorrelation loss, iterative normalization **(Huang et al., 2019)**, and expert yielded estimates (EYE) regularization **(Wang et al., 2022)**.

* undefined cross-reference in Appendix E.1: "*Section??*"
* The description of CUB in Section 6 should state that the concept set of Koh et al. (2020) is used, because the original CUB dataset has 312 attributes.